# Generalist species drive microbial dispersion and evolution

Sira Sriswasdi [1,2], Ching-chia Yang [1] & Wataru Iwasaki [1,3,4]

Microbes form fundamental bases of every Earth ecosystem. As their key survival strategies, some microbes adapt to broad ranges of environments, while others specialize to certain habitats. While ecological roles and properties of such "generalists" and "specialists" had been examined in individual ecosystems, general principles that govern their distribution patterns and evolutionary processes have not been characterized. Here, we thoroughly identified microbial generalists and specialists across 61 environments via meta-analysis of community sequencing data sets and reconstructed their evolutionary histories across diverse microbial groups. This revealed that generalist lineages possess 19-fold higher speciation rates and significant persistence advantage over specialists. Yet, we also detected three-fold more frequent generalist-to-specialist transformations than the reverse transformations. These results support a model of microbial evolution in which generalists play key roles in introducing new species and maintaining taxonomic diversity.

---

[1] Department of Biological Sciences, Graduate School of Science, the University of Tokyo, Bunkyo-ku, Tokyo 113-0032, Japan. [2] Research Affairs, Faculty of Medicine, Chulalongkorn University, Pathum WanBangkok 10330, Thailand. [3] Department of Computational Biology and Medical Sciences, Graduate School of Frontier Sciences, the University of Tokyo, Kashiwa, Chiba 277-8568, Japan. [4] Atmosphere and Ocean Research Institute, the University of Tokyo, Kashiwa, Chiba 277-8564, Japan. Correspondence and requests for materials should be addressed to S.S. (email: sira.sr@chula.ac.th) or to W.I. (email: iwasaki@bs.s.u-tokyo.ac.jp)

Microbes exist almost everywhere on Earth, forming fundamental bases of every ecosystem[1]. The large population size and small size of microbes enable them to move across and colonize diverse environments[2–5]. In response to constant movements and competitions against invading species, by becoming generalists (i.e., those that are able to adapt to diverse habitats) or specialists (i.e., those that adapted to specific habitats), microbes improve their survivability[6,7]. Previous works have shown that these generalist and specialist microbes differently impact the dynamics of microbial community structures[8,9]. However, the mechanisms behind the evolution and dispersion of these microbes are not well understood, especially given that the evolution of specialization in macro-organisms has been investigated in many phylogenetic groups[10,11]. Furthermore, while recent advances in high-throughput sequencing have enabled in-depth analyses of individual microbial ecosystems[12], little is known about the general, global principles that govern the dynamics of microbial communities and gene pools[13,14].

In this study, we performed a large-scale meta-analysis of community sequencing data sets that sheds light on the distribution and evolutionary history of generalist and specialist microbes. Generalist microbes were found to have significantly higher speciation rates and persistence advantages over their specialist counterparts. Rapid generalist-to-specialist transformation rates and a positive correlation between the presence of generalists and increased habitat diversity of their close specialist relatives suggest that descendants of generalist lineages evolve into new specialist species across diverse environments. Collectively, our findings highlight key evolutionary and ecological roles of generalist species.

## Results

**Meta-analysis of community sequencing data sets**. To characterize generalist and specialist microbes on a global scale, and to elucidate their impacts on microbial community and dispersion, we applied a maximum likelihood binary-state model

approach[15,16] to 16S ribosomal RNA (rRNA) sequence data sets from 61 environments, ranging from host-associated environments such as human respiratory tract, human gut, and insects to marine, soil, freshwater, and bioreactor (Supplementary Data 1). Community sequencing data were collected using MetaMetaDB[17], a database for meta-analysis of metagenomic and 16S rRNA sequence data sets, which identified ~2.7 million 16S rRNA fragments. "Singleton" fragments that had no similar sequence (≥98% sequence identity) were discarded, as they may result from erroneous sequencing or prediction. The remaining fragments were mapped to non-redundant full-length 16S rRNA sequences in the SILVA database[18] (Methods; Fig. 1a; Supplementary Table 1), whereas fragments that could not be mapped were removed from further analyses. When a 16S rRNA fragment could be mapped to multiple full-length sequences with high confidence, that fragment was assigned to the candidate with the most fragment hits in accordance with the Occam's razor principle. Then, each full-length sequence was assigned to environments that were supported by at least 10 mapped fragments (Fig. 1b; Supplementary Data 2). Read counts of the predicted 16S rRNA fragments were also combined together to estimate the abundance of each mapped full-length sequence (Methods).

To improve the robustness of environment assignment and to alleviate redundancies in the 61 environment annotations in the original database, we grouped environments that contain similar profiles of the assigned full-length 16S rRNA sequences (Fig. 1c). Specifically, we defined an environmental similarity score between two environments as the geometric mean of the proportions of the assigned 16S rRNA sequences in one environment that had close relatives (≥98% sequence identity) in the other (Methods). To facilitate inference of congruent habitat associations and life styles, we selected the 98% threshold that is more stringent than the more commonly used threshold of 97% for defining microbial species[19,20]. Agglomerative hierarchical clustering guided by this score revealed 11 major environment clusters that remained quite consistent upon changing the sequence identity threshold for calculating the

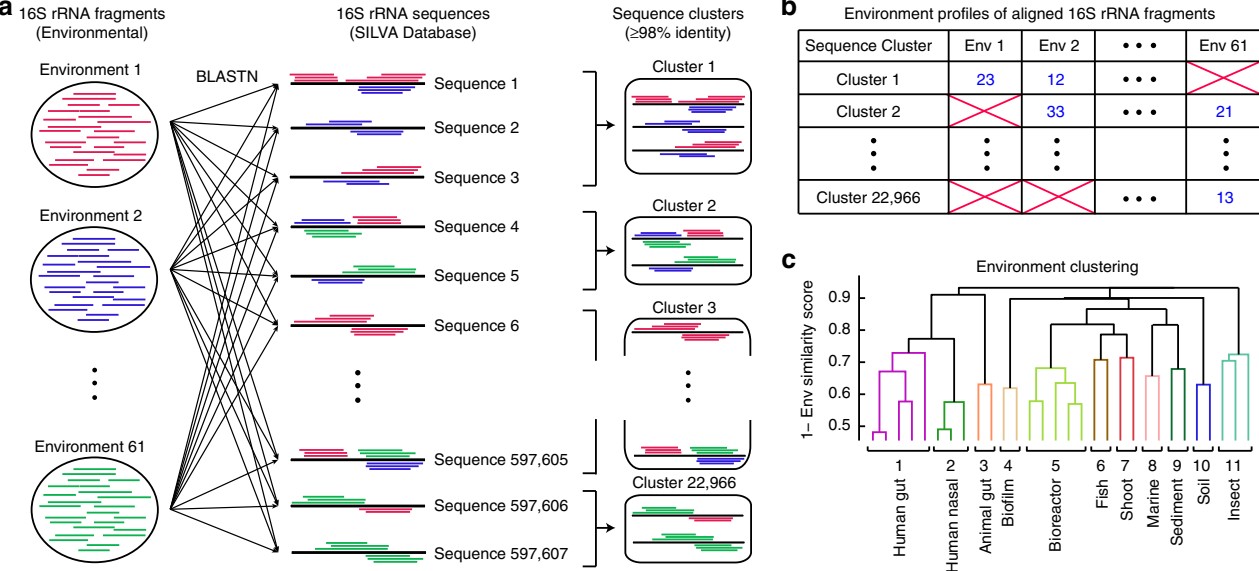

**Fig. 1** Clustering similar environments based on 16S rRNA sequence profiles. **a** Predicted 16S rRNA fragments from community sequencing data sets were mapped to full-length 16S rRNA sequences from SILVA database at 98% identity threshold. Redundant full-length sequences were then clustered at 98% identity threshold. **b** Full-length 16S rRNA sequences were assigned to environments based on mapped fragments. **c** Environments were clustered based on the similarity in their full-length 16S rRNA sequence profiles (Methods). Eleven non-singleton clusters were identified

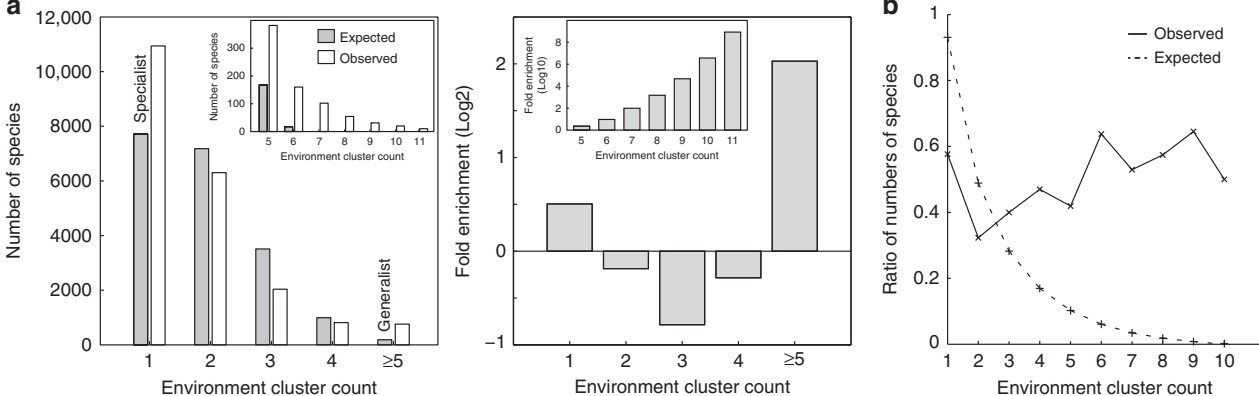

**Fig. 2** Classification of generalists and specialists. **a** The distribution of the number of environment clusters that a species belongs to was compared to the expected distribution derived from 100,000 permutations. The insets show the detailed histograms for species that belong to five or more environment clusters. **b** The rate of new habitat acquisition (the y-axis, "Ratio of Number of Species"), as represented by ratio of the number of species that belong to $N + 1$ environment clusters over the number of species that belong to $N$ clusters, are higher than expected, especially for classified generalists (species that belong to at least five environment clusters). The x-axis indicates the number of clusters, $N$

similarity score and when the 16S rRNA fragments were directly analyzed instead of the full-length sequences (Supplementary Fig. 1). Each of the 11 environment clusters contained related environments (e.g., soil and rhizosphere in one cluster and human lung, nasal pharyngeal, and oral in another cluster) with similar species abundance distributions (Supplementary Fig. 2). These findings reflect the observations that microbes are not neutrally distributed on such a broad scale of environment definition[21].

**Distribution of generalist and specialist microbes**. Next, generalist and specialist species were identified from the species–environment association pattern. Comparison of the distribution of the number of environment clusters associated with each species to the expectation derived from 100,000 permutations (Methods) showed significant enrichment of species that belong to exactly one cluster and of species that belong to five or more clusters (Fig. 2a, permutation test p-values < 1e−5). This led us to classify 9464 species that belong to one cluster as specialists and 759 species that belong to five or more clusters as generalists (Supplementary Data 3). Another distinctive discovery concerns the rate of new habitat acquisition, as approximated by the ratios between numbers of species that belong to $N + 1$ clusters and number of species that belong to $N$ clusters. This rate monotonically decreases with increasing number of environments in the expected distribution (Fig. 2b), i.e., the null hypothesis of random association between species and environments predicts that it is progressively unlikely for generalist species to acquire additional habitat. However, the trend is reversed in the observed distribution (Fig. 2b, Spearman rank correlation = 0.7667 with permutation test p-value = 0.012 for $N > 1$), pointing to increased capability of generalist species to acquire more habitats. Analysis of genomes and predicted proteomes downloaded from NCBI's reference genome database revealed that generalists tend to possess larger genomes and proteomes (sign test p-values for the log ratios of generalist-to-specialist genome and proteome sizes = 1.94e−11 and 2.13e−4, respectively, Methods). This is in good agreement with the expectation that generalists require larger gene and protein repertoires to survive in multiple environment conditions. On the other hand, we did not find other genomic characteristics that significantly differ between generalists and specialists (e.g., for genomic G + C content, the sign test p-value is 0.4582).

From a taxonomic perspective, it is notable that distributions of numbers of habitat per species differ sharply at such broad scale as major bacterial phyla (Supplementary Table 2). In particular, Proteobacteria and Bacteroidetes show contrasting trends by having significantly more generalists and specialists, respectively. This pattern coincides with the fact that most members of Proteobacteria possess complete flagellar system (16.48 out of 21 core flagellar genes[22] on average, Methods), while few members of Bacteroidetes do (1.69 out of 21 core flagellar genes on average). Such widespread flagellar motility among Proteobacteria, in conjunction with their metabolic versatility[23], may contribute to their high number of habitats. Nonetheless, enrichments of generalists or specialists across microbial phyla are likely to be driven by a number of different molecular bases. At lower taxonomic levels, groups that were most significantly biased toward generalists were genus *Mycobacterium* and family Streptococcaceae, both of which were known to contain free-living, widespread species as well as obligate pathogens[24]. On the other hand, groups most biased toward specialists consisted primarily of obligate anaerobes such as the class Clostridia, family Lachnospiraceae, genus *Christensenella*, and genus *Prevotella*[25,26].

**Evolutionary characteristics of generalists and specialists**. To investigate whether being a generalist or specialist affects the evolutionary characteristics of a species, we defined being a generalist and specialist as distinct evolutionary states according to the Binary-State Speciation and Extinction (BiSSE) model[15] (Fig. 3a). The generalist and specialist species were mapped (≥98% sequence identity) to the archaeal-bacterial phylogenetic tree obtained from the SILVA Living Tree Project (LTP)[27,28]. Then, the subtree consisting of 1255 mapped leaf nodes (710 generalists and 545 specialists, Supplementary Data 4) was linearized using a divide-and-conquer approach (Methods, Supplementary Fig. 3). Finally, the evolutionary rate parameters of the BiSSE model (i.e., speciation, extinction, and state-transition rates for both states) were estimated with a maximum likelihood method[16]. As the algorithm for solving the BiSSE model heavily relies on given phylogeny, we repeated the rate estimation under various linearized phylogenetic trees obtained by either altering branch lengths or randomly removing leaves and confirmed that the estimated rates are robust (Supplementary Fig. 3). This revealed that generalists possess 19-fold higher speciation rates than specialists do (Fig. 3b;

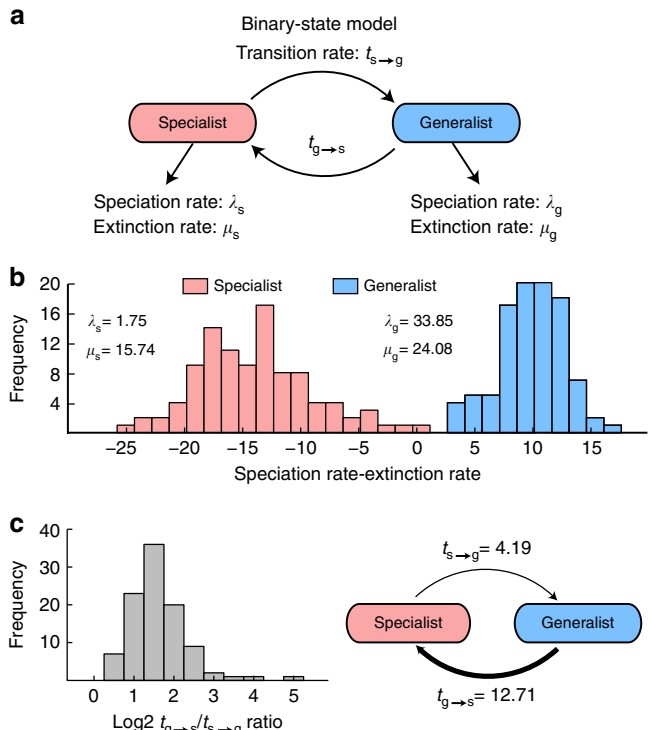

**Fig. 3** Estimation of evolutionary characteristics for generalists and specialists. **a** Binary-state speciation and extinction (BiSSE) model for the evolution of specialists and generalists. Each state has distinct speciation ($\lambda$), extinction ($\mu$), and state-transition ($t$) rates. **b** Generalists possess 19-fold higher speciation rates than specialists do. The net expansion rates (the differences between speciation and extinction rates) are positive for generalists and negative for specialists. Histogram data comes from 100 random subsamples (Methods). Average speciation and extinction rates for each state are indicated. **c** Generalists transform into specialists at three-fold faster rates than the reverse transitions. Histogram data come from 100 random subsamples (Methods). Average transition rates are indicated

Supplementary Fig. 3). Because a microbe's proteome can be substantially affected by environmental conditions[29], the elevated speciation rates could be due to generalists' exposure to diverse environments each requiring a distinct gene set for survival and each exerting different evolutionary pressures. Specialists, by similar arguments, may exhibit reduced speciation rates due to their limited ability to spread into new habitats[30].

Despite our findings that being a generalist confers evolutionary advantages, with significant and positive expansion rates (Fig. 3b, the difference between speciation and extinction rates), the number of detected specialists still far outnumbers that of generalists (e.g., here and ref. [31]). This is likely due to rapid transformation of generalists into specialists, which is three-fold more frequent than the reverse transitions (Fig. 3c; Supplementary Fig. 3). The imbalance in transition rates might result from the evolutionary pressures that drive descendants of a generalist lineage to become more efficient in their new habitats, as well as to lose extraneous genes from their genomes[32,33]—essentially turning them into specialist species. As a consequence of such reductive evolution, most of the older species already became specialists and present-day generalists consist of significantly younger species (Mann–Whitney $U$-test $p$-value for the lengths of phylogenetic branches directly leading to generalist and specialist species = 9.87e−52). These findings suggest that being a generalist is a relatively transient evolutionary state and most species will eventually turn into specialists.

**Robustness of estimated evolutionary characteristics**. To evaluate the robustness of our findings, we first re-performed all analyses using a less stringent sequence identity threshold of 95%. No major change in the results was observed (Supplementary Fig. 1 for the clustering of environments, and Supplementary Fig. 4 for the estimation of evolutionary rates). At both 98 and 95% sequence identity thresholds, generalists consistently exhibited more than 10-fold higher speciation rates than specialists did. The net expansion rates (the difference between speciation and extinction rates) were positive for generalists and negative for specialists. The generalist-to-specialist transformation rates were significantly higher (three-fold or more) than the reverse transitions. This indicated that these findings are not sensitive to the definition of microbial species and that similar findings may be found at broader taxonomic classification levels. Next, we investigated two possible scenarios that could affect the classification of generalists and specialists (Supplementary Fig. 5a, b). In the first scenario, a rare species may be detected in only one environment simply because it is rare and erroneously classified as a specialist. In the second scenario, species that are present in multiple environments but are abundant in only one environment may be classified as a generalist even though it resembles a specialist. That is, classified specialists with the lowest abundances and classified generalists whose abundance profiles have the lowest Shannon entropy indices are expected to be the most susceptible to misclassification. However, minor impacts of the removal of these species on the estimated evolutionary characteristics (Supplementary Fig. 5c) suggested that the classification of generalists and specialists based on the numbers of environment clusters was robust to misclassification.

We further examined an alternative method for classifying generalists and specialists that is based on comparing the numbers of environment clusters in which a species was found against the expectation given that species' occurrence pattern, thereby reducing the impact of species abundance on classification (Methods, Supplementary Fig. 6a). To compare to our original classification, the thresholds on the difference between the observed and expected numbers of environment clusters were selected so that both methods produce the same numbers of classified generalists and specialists (Supplementary Fig. 6b). Although the overlaps were modest (740 out of 1603 mapped leaves at the mapped-phylogenetic-leaf level), estimation of the evolutionary characteristics revealed that all key findings—namely, 10-fold higher speciation rates for generalists than specialists, positive net expansion rates for generalists and negative for specialists, and significantly higher generalist-to-specialist transition rates than the reverse transition—remained consistent across the two classification schemes (Supplementary Fig. 6c).

## Discussion

The evolutionary characteristics of generalists and specialists suggest a model of microbial dispersion driven by the ability of generalist species to expand across ecosystems (Fig. 4a–f). Once a generalist species emerges (Fig. 4b), its descendants then spread into new environments (Fig. 4c), experience distinct evolutionary pressure, and evolve into new species (Fig. 4d). Eventually, these species specialize to their new habitats (Fig. 4e). Each round of such generalist–specialist evolutionary cycle would introduce a larger number of new specialist species across diverse environments that can shift the microbial community structure (Fig. 4f). In fact, we found that when a microbial clade contains high frequency of generalist species, it also tends to contain specialist species that belong to diverse environments (Fig. 4g, Spearman correlation between the frequency of generalist and normalized entropy of habitat diversity of specialists = 0.4281 with

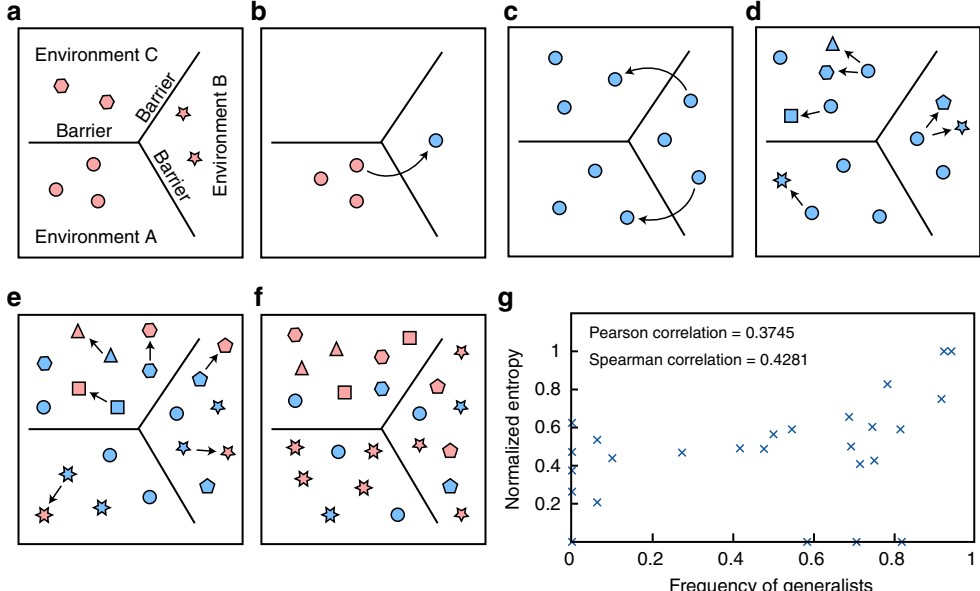

**Fig. 4** The model of microbial dispersion and evolution driven by generalist species. Generalists and specialists are represented in blue and red, respectively. Each shape represents a species. **a** Habitat occupancy of specialists is limited by environmental barriers. **b** A new generalist seldom emerges from a specialist lineage. **c** Descendants of the new generalist lineage expand into new environments. **d** Each environment separately drives the evolution of each descendant. **e** Eventually, some descendants begin to specialize to their new habitats. **f** This cycle of transformations effectively introduces a large number of new species across multiple environments. **g** Scatterplot between the frequency of generalist species and the normalized entropy of the distribution of specialist species among 11 environmental clusters. Each data point corresponds to one of the partition of the archaeal-bacterial phylogenetic tree (Methods). High normalized entropy means that specialist species in that cluster belong to diverse habitats

permutation test $p$-value = 0.015). It should be noted that our model is in good agreement with the theory of adaptive radiation[34,35] in that it highlights both the early burst of speciations in the ancestral generalist lineage and the transition to low speciation rates and high extinction rates in specialist descendants. Furthermore, the generalist–specialist evolutionary cycle implies that microbial diversity is maintained more through continual replenishments of specialists from generalist lineages than through continued persistence of specialist lineages[35].

Interestingly, past studies of generalist and specialist macroorganisms turned up inconsistent conclusions regarding the evolutionary consequences of specialization[10,11]. Depending on the phylogenetic groups being studied, generalists could be shown to possess higher or lower speciation rates than specialists do. In one case, a change in analysis methodology even reversed the original finding that generalists evolve from specialists[11,36]. While small numbers of analyzed species or strong biases toward generalists or specialists could be a reason behind the discrepancy[37], our data set does not fall into these scenarios. Another key complication in these studies lies in how they define generalists and specialists, since a species can be considered a specialist in one aspect while being a generalist in another[10]. Differences in the population dynamics and reproductive systems between micro- and macroorganisms could also lead to divergent evolutionary outcomes of generalization and specialization. For example, for sexually reproduced macroorganisms, high dispersibility of generalists may end up suppressing their speciation rates by increasing gene flow and reducing the likelihood of reproductive isolation[38]. Finally, it is notable that at least one evolutionary consequence of specialization is clear across micro- and macroorganisms. With smaller habitat ranges and greater reliance on specific resources, specialist species have higher chances to become extinct[10,38].

In summary, integrative analysis of microbial habitat profile and evolutionary history permitted identification of generalist and evolutionary microbes across the tree of life and characterization

of their evolutionary impacts. Our results complement other studies of microbial dispersion and evolution of specialization by presenting a general model of how microbial species flow across environments and highlighting generalist species as key players in this process. Furthermore, our approach exemplifies the notion that current microbial community structures are shaped by not only contemporary factors but also historical and evolutionary events[39]. We anticipate that the generalist-driven evolutionary cycle may represent a fundamental force that underlies the highly dynamic microbial gene pools[14] and the high mobility of antibiotic resistance and viral genes[40].

## Methods

**Microbial community sequencing data sets**. Microbial community sequencing data sets were downloaded from the DDBJ Sequence Read Archive (DRA)[17] in March 2014 (Supplementary Data 1). Reads were processed step-by-step as described below. First, TrimmingReads.pl script of NGS QC Toolkit[41] was used to trim low-quality reads with Phred-equivalent quality scores of below 20. Cutadapt version 1.1[42] was then used to remove adapter sequences. Ambiguous reads and homopolymers consisting of five or more base pairs (bp) were removed using AmbiguityFiltering.pl and HomopolymerTrimming.pl scripts of NGS QC Toolkit, respectively. CD-HIT-454[43] was used to remove possible artificial duplicates at a 99% threshold. At each step, short sequences (<200 bp) were removed. Finally, 16S rRNA sequences were predicted using SortMeRNA version 1.8[44] with the non-redundant SILVA database[18] as reference. UCHIME[45] was used to remove possible chimera sequences. In total, our processed data set contained 2,737,833 predicted 16S rRNA sequence fragments with an average length of 362 bp. The list of processed 16S rRNA sequences can be found on MetaMetaDB's server[46] at http://mmdb.aori.u-tokyo.ac.jp/archive.html.

**Mapping of 16S rRNA fragments to full-length sequences**. To associate predicted 16S rRNA sequence fragments to full-length sequences, a two-step BLASTN was performed using BLAST+ version 2.2.26+[47]. First, an all-against-all BLASTN of the predicted sequences was performed to cluster sequences with high identity (identity ≥ 98%, coverage ≥ 70%, and E-value ≤ 1e−5). This produced 132,927 clusters that contain multiple sequences with an average of 7–8 sequences per cluster. Singleton clusters, which accounted for ~60% of the fragments in the data set, were removed from consideration as they may represent erroneous sequences or predictions. For each cluster, the sequence with the highest average identity to the other members of the same cluster was chosen as the representative

sequence. Second, the representative sequences were searched against the non-redundant SILVA database to group sequences that correspond to different parts of the same full-length 16S rRNA sequence (identity ≥ 98%, coverage ≥ 70%, and E-value ≤ 1e−5). When a predicted sequence could be matched to multiple full-length 16S rRNA sequences, we applied the Occum's Razor principle and assigned that predicted sequence to the full-length sequence that contained the most BLASTN hits. Approximately 40% of the predicted fragment clusters were mapped to some full-length sequences (Supplementary Table 2). To remove redundant SILVA entries, we clustered the full-length sequences and merged their mapped fragments (identity ≥ 98%, coverage ≥ 70%, and E-value ≤ 1e−5). Finally, we assigned each full-length sequence cluster to environments that was supported by at least 10 mapped fragments. This yielded a species–environment association table consisting of 21,345 full-length sequence clusters (Supplementary Table 3).

**16S rRNA abundance estimation.** We estimated the abundance of species associated with each non-redundant full-length 16S rRNA sequence across all environments using the sequence read counts of predicted 16S rRNA fragments. Due to the fact that our data set consists of a large number of sequencing experiments submitted by different researchers and performed on different platforms, we opted to estimate species abundance on a relative scale rather than an absolute scale. Specifically, for each full-length sequence, its relative abundance in an environment is defined as the fraction of the total 16S rRNA fragment read of that environment that was mapped to that full-length sequence. This relative abundance was used to sort classified specialists to identify rare species and for the calculation of Shannon entropy for classified generalists. It should be noted that the total 16S rRNA fragment read (i.e., the denominator of the fraction) includes singleton fragments and fragments that could not be mapped to any SILVA entry since they are also a part of the 16S rRNA content of that environment.

Then, to account for the fact that different environments have different proportions of unmapped fragments and different numbers of detected species, both of which would influence the range of estimated relative abundances, we converted relative abundances into rank-based percentile scores. In each environment, all member species were sorted by their relative abundances and assigned "abundance percentile scores" based on their percentile ranks. This enabled fair comparisons of species abundances across multiple environments (Supplementary Fig. 2)

**Environmental similarity score and clustering.** Given two environments $E_1$ and $E_2$ and a sequence identity threshold $T$ ($T = 98\%$ for the main results), we defined a similarity score between $E_1$ and $E_2$ as the geometric mean of (i) the proportion of 16S rRNA sequences in $E_1$ that was similar to any sequence in $E_2$ at the identity threshold $T$ and (ii) the proportion of 16S rRNA sequences in $E_2$ that was similar to any sequence in $E_1$ at the identity threshold $T$ (Fig. 1b). Hierarchical clustering of the 61 environments was then performed using the dissimilarity score, or 1—similarity score, and the Weighted Pair Group Method with Arithmetic Mean (WPGMA) method. This identified 11 non-singleton environment clusters that are quite consistent upon changing the sequence identity threshold or replacing full-length sequences with predicted fragments (Fig. 1c; Supplementary Fig. 1). For each pair of environments, we also calculated the Spearman rank correlation between the abundance profiles of species that were detected in both environments. We found that the species abundance profiles of environments within the same cluster are significantly more positively correlated than those of environments from different cluster (Supplementary Fig. 2, Mann–Whitney U-test p-value = 8.9e−4).

**Enrichment analysis of habitat specialists and generalists.** To identify generalists and specialists, we first computed the enrichment of the number of full-length 16S rRNA sequence clusters (regarded as species from here on) that were assigned to a particular number of environment clusters. We performed 100,000 random permutations of the species–environment association map, in such a way that the cluster sizes are always preserved, to obtain the background distribution. This revealed that there are enrichments of species that belong to a single cluster and those that belong to five or more clusters (Fig. 2a). Therefore, we classified species that belong to a single cluster as "specialists" and those that belong to five or more clusters as "generalists". In total, there were 759 generalist species and 9464 specialist species (Supplementary Table 4).

**Archaeal-bacterial phylogenetic tree.** The pre-constructed phylogenetic tree of microbial small subunit (SSU) rRNA sequences was downloaded from SILVA's All-Species Living Tree project[27,28] (LTP, release 123). The generalist and specialist species identified above were mapped to this tree using BLASTN (identity ≥ 98%, coverage ≥ 70%, and E-value ≤ 1e−5). Reciprocal best hits were assigned first; the other hits were then assigned in the order from high- to low-sequence identity. This let us label 710 leaves of the LTP tree as generalist and 545 leaves as specialists (Supplementary Table 5).

To prepare for subsequent analyses of evolutionary characteristics of generalists and specialists, we extracted a subtree of LTP tree consisting 1255 mapped leaves and linearized it. As this tree was large and contained distantly related species, we applied a divide-and-conquer strategy to facilitate the linearization. First, the tree was split into partitions based on the distribution of generalists and specialists.

Then, individual partitions were linearized separately and joined back together according to the topology of the LTP tree (Supplementary Fig. 3). The partition algorithm was as follows: starting from the root of the tree and traversing down the tree in breadth-first order, at each internal node, the proportions of generalist species in its left and right child subtrees were calculated. If the two proportions differ by more than 0.1, then the tree was split at this internal node. To prevent erroneous partitioning due to the discreteness of proportion estimates, we stopped splitting the tree if it would create a partition with less than six species and later discarded small partitions that contained <10 species each. This resulted in 32 partitions covering 622 species (399 generalists and 223 specialists, the "Partition ID" column of Supplementary Table 5).

For each partition, 16S rRNA sequences of all involved species were aligned to the reference alignment of every non-redundant SSU rRNA sequence (SILVA release 123, version 12/07/15) using SILVA Incremental Aligner (SINA) version 1.2.11[48]. The linearized phylogenetic tree was then reconstructed using the baseml module of PAML version 4.8[49] with molecular clock option. The input tree topology was fixed according to the LTP tree. Generalized time-reversible (GTR, also known as REV in PAML) substitution model was used. Then, all linearized trees were joined according to the topology of the LTP tree in such a way that all ancestral branches (those not directly connected to the linearized trees) were set to length $\alpha$; the lengths of branches directly connected to the linearized trees were set so that all root-to-leave distances are identical (Supplementary Fig. 3). Because the root-to-leaf distances of the linearized trees range from 0.021 to 0.419, we tried different values of the ancestral branch length $\alpha$, ranging from 0.2 to 5.0, to investigate its impact on subsequent analyses (Supplementary Fig. 3).

**Binary-state speciation and extinction model analysis.** To determine the evolutionary characteristics of generalists and specialists, we utilized a binary-state model that includes two different speciation, extinction, and state-transition rates (Fig. 3a). The model was termed Binary-State Speciation and Extinction (BiSSE)[15] and implemented in an R package[16]. For each input linearized phylogenetic tree, diversitree was run twice: once to determine a starting point for the simulation, and once more to determine the maximum likelihood estimate for the rate parameters. During the first round, the two states were constrained to have an identical speciation rate and an identical extinction rate. This was simply to improve upon the starting point provided by the package's *starting.point.bisse* function. The second round was run with all rates allowed to be different. Afterward, 1000-step Markov Chain Monte Carlo (MCMC) simulations were performed using the package's *mcmc* function to assess the stability of the final estimation. Furthermore, because the implementation of BiSSE model in diversitree assumes that the input phylogenetic tree contains all surviving species, which is not the case here, we examined the impacts of such missing information as well. We generated 100 subsamples of the linearized phylogenetic tree at 0.8 sampling rate and analyzed them with diversitree. This revealed that the estimated rates are quite consistent (Supplementary Fig. 3).

**Genome statistics.** Genome and proteome sequences were extracted from NCBI's reference genome database in July 2016. The list of all bacterial genomes was obtained from NCBI's Reference Sequence collection[50]. To map between SILVA entries and NCBI's genome assembly accession numbers, we extracted species name from SILVA database (Supplementary Table 4) and matched them to organism names in NCBI's assembly summary file (ftp.ncbi.nlm.nih.gov/genomes/refseq/bacteria/assembly_summary.txt). Only exact matches were considered. We also made sure to exclude uninformative terms such as "uncultured archaeon" from consideration. When multiple genome assemblies of the same organism were present, all assemblies were downloaded. Predicted gene and protein sequences were also downloaded from NCBI when they were present. Ambiguous base calls that were neither A, T, C, nor G were excluded from all calculations of genome sizes and G + C content. Proteome size was calculated as the total number of amino-acid residues of the predicted proteome (downloaded from NCBI). For species with multiple genome assemblies, the average statistics were taken. To control for phylogenetic effects when comparing genome statistics across generalist and specialist species, we only considered pairs of generalist and specialist whose 16S rRNA sequences were at least 85% identical. Specifically, for each generalist, we searched for specialists using an 85% identity threshold on 16S rRNA sequences. In total, 818 such generalist–specialist pairs were found between 225 generalists and 113 specialists. For each pair, we calculated the ratios of genome size, proteome size, and G + C contents between the generalist and the specialist. This revealed that the genome and proteome sizes are significantly larger in generalists (sign test p-values of the log of ratios = 1.94e−11 and 2.13e−4, respectively) whereas the G + C content does not show any significant difference (sign test p-value of the log of ratios = 0.4582).

**Characteristics of the linearized phylogenetic tree.** As a confirmation for the modeling results, we examined the linearized phylogenetic tree directly. To evaluate the age of currently existing generalist and specialist species, the lengths of phylogenetic tree branches directly leading to these species were compared. This revealed that generalists are significantly younger (Mann–Whitney U-test p-value = 9.87e−52, three-fold difference in average branch lengths). Also, for each partition of the tree defined in the section "linearizing phylogenetic tree", we calculated

the proportion of generalist species and the habitat distribution of specialist species along with its normalized entropy. This revealed that the proportion of generalist positively and significantly correlates with the normalized entropy (Fig. 4g, Spearman correlation = 0.4281 with permutation test p-value = 0.015). Permutation tests were carried out by randomly shuffling the original data 10,000 times. For the calculation of specialist's habitat profile and its normalized entropy, only partitions containing multiple specialist species were considered (26 out of 32 partitions).

**Detection of flagellar genes**. The list of Proteobacteria and Bacteroidetes species were extracted from SILVA non-redundant database (1248 Bacteroidetes and 4413 Proteobacteria). Their annotated proteomes were then downloaded from NCBI (480 Bacteroidetes and 1634 Proteobacteria). When proteomes for multiple strains of the same species are present, only the first entry as listed in NCBI's bacterial assembly summary (ftp.ncbi.nlm.nih.gov/genomes/genbank/bacteria/assembly summary.txt, January 2017 version) was picked. The amino-acid sequences of 21 core flagellar proteins[22] were extracted from *Escherichia coli* genome. BLASTP was performed with the core proteins as query and each of the 2114 proteomes as database. A flagellar protein is considered present in a proteome if there was at least one BLASTP hit with E-value < 1e−5.

**Alternative classification**. Each species' occurrence profile was randomly permuted 10,000 times and the corresponding numbers of environment clusters were calculated (Supplementary Fig. 6a). The difference between the observed and expected numbers of environment clusters over all permutations was then used as the criterion for classifying species as generalists (observed number is larger) or specialists (expected number is larger). To compare this permutation-based method to the number-of-clusters-based method described above, we set thresholds on the difference between the observed and expected numbers of environment clusters so that the same numbers of classified generalists and specialists were obtained (Supplementary Fig. 6b). Then, evolutionary characteristics for the generalists and specialists classified with this alternative method were estimated (Supplementary Fig. 6c).

**Data availability**. Predicted 16S rRNA sequences have been deposited in MetaMetaDB's server[46] at the University of Tokyo (http://mmdb.aori.u-tokyo.ac.jp/archive.html). The authors declare that all other data supporting the findings of this study are available in the manuscript and its supplementary files or are available from the corresponding author upon request.

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

## Acknowledgements

This work was supported by the Japan Society for the Promotion of Science (Grant Numbers 14F04382 and 16S06154), the Ministry of Education, Culture, Sports, Science and Technology in Japan (221S0002 and 16H06279), the Japan Science and Technology Agency (CREST), and the Canon Foundation. We would like to thank Tomoyuki Mikami, Seishiro Aoki, and Sohta Ishikawa for their contributions to improving the analyses and discussion on evolution and ecology.

## Author contributions

S.S. and C.-c.Y. performed data analyses. S.S., C.-c.Y., and W.I. wrote the manuscript. W.I. directed and supervised the research.

## Additional information

**Competing interests:** The authors declare no competing financial interests.

