## [Peer Review File · Nature Communications]

1st Round of Review

Reviewer #1 (Remarks to the Author):

The manuscript reports a massive analysis of the distribution of specialization of microbes. Specialization is measured as the number of 'environments' in which a given species is found. The main result is that generalists do have a much larger diversification rate than specialists. Based on this result, the authors develop the hypothesis that microbial evolution happens as a cycle of generalization and specialization.

I found the results very intriguing, and if they prove solid, I think it would make a very significant contribution to ecology and evolution, beyond microbial ecology. That said, I must admit that I am marginally qualified to judge the methodology that is employed in this study, and for most of what will follow I will assume that the technics are good. For me the most interesting part is the evolution cycle, which is quite contrary to convention with other organisms. The problem of the evolution of specialization has been investigated quite a lot, as well as the evolution of niches. Unfortunately, the authors do not place their results in this context and stay at the level of microbiology. I would encourage them to visit some of this literature and place their results in a wider context, asking if microbes are exceptional or coherent with other hypotheses (from what I see, these results are quite exceptional).

While I am overall very positive about the paper, I have one general comment that might at the end be a major limitation preventing publication : the conclusions critically depend on how generalization / specialization is measured for each lineage. The authors cluster the usage of different environments, and then compute for each lineage the number of clusters it is found at. According to this approach, lineages that are qualified as generalists can't be otherwise, they are found at several places. But the reverse is not necessarily true, as a species could be qualified as specialist only because it is rare. The odds of a rare lineage be found at several locations are much smaller, up to the point where a lineage observed only twice could not be found at more than two locations. Consequently, there is no guarantee for now that lineages qualified as generalists are true generalists, even with the null model analysis performed.

There is at least one way to check if this is problematic or not : the authors should look at the relationship between the number of times of a given species is observed against the number of habitats it is found at. If the relationship is triangular, then there is a big problem. In such case, the analysis could not discriminate between the effect of the strategy and of abundance.

I could see two ways to bypass this problem :

- first, instead of using binary descriptors of the strategy, the authors could use a continuous measure of specialization. There are tons of them, but one that is easy to perform is simply the Shannon entropy computed on the frequency of different environments for a given taxa. This approach would not completely solve the problem (would be hard to compute on rare species), but at least it would account for the fact that a species might have been found at 5 different environments, but 99% of its presence is at a single environment (this species would then qualify as a specialist, despite being found everywhere).

- The other option is to perform the opposite of rarefaction analysis and compute the asymptotic specialization as sequences are removed. This way, the measure would standardize the number of occurrences.

Other than type, I would propose extra testable predictions to test the evolution cycle hypothesis:

- If the hypothesis is right, then two sister species would be less likely to have the same strategy (generalist/specialist) than two species picked at random (also called overdispersion of strategies)

- The distribution of preferred environments should however be clustered (sister species are more likely to be found at the same environment)

I have very few minor points, the manuscript was well written and prepared. I really enjoyed reading it.

- Definition of specialisation should come early in the manuscript

- L22 : replace 'survival' by 'persistence'

- L43 : it would be useful to give some ideas of the types of environments that are documented in the main text

- L45 : replace 'predicted' by collected

- L46: what is the fraction of singletons ?

- Some discussion on the sensitivity of the analysis to the definition of what constitutes a species should be provided.

I signed my evaluation

Dominique Gravel

Reviewer #2 (Remarks to the Author):

The paper investigates the dynamics of diversification of microbial systems across a range of environments. The authors identify some unexpected patterns in the evolutionary dynamics of specialist and generalist microbial species. Overall this study is quite thorough, the results are extremely interesting and important. They also appear to be solid, and the writing style is good.

- What I find missing are any links to speciation/adaptive radiation literature/thinking in evolutionary biology. This is a shame because this paper has a potential to greatly contribute to ongoing discussions of general patterns of biological diversification in evolutionary biology (e.g., Gavrillets and Losos 2009, Science 323:732-737).

- Birand et al (2012, Am. Nat. 79:1-21) studied speciation dynamics within a generalist-specialist framework. Some of their theoretical predictions should be contrasted with the authors' results.

- Also, the title doesn't really reflect what this paper is about.

Reviewer #3 (Remarks to the Author):

The manuscript by Sriswasdi et al. describes a meta-analysis of existing 16S rRNA data from different environments with the aim of inferring survival strategy (generalists vs. specialists) and linking these to different evolutionary patterns. The inference of microbial lifestyle strategies from omics data is an emerging area in microbial ecology and so far rather limited work has been done on translating these concepts from macroecology into microecology (but see Muller et al., 2014, Nature Communications 5, 5603). Although this paper tackles important questions, I am presently unconvinced about the approach given major flaws in the author's interpretation of what generalists and specialists represent. Please find my detailed comments below.

Major concerns

1. The use of the terms generalist and specialist as well as the linked concepts are used incorrect. A generalist is able to thrive in a wide variety of environmental conditions and can utilise a variety of different resources – a generalist has a broad ecological niche. In contrast, a specialist can only thrive in a narrow range of environmental conditions or has a limited diet. As the authors do not take into account population sizes, environmental conditions in the different environments they are looking at, different resource usage patterns by the different species, etc. they are unable to discern between generalists and specialists. In fact, from their analyses, the authors are able to differentiate between cosmopolitan and endemic species but are clearly unable to infer survival strategy. This misinterpretation of the data results in all kinds of issues for example the much higher speciation rate for “generalists” which is completely counter to what has been described for macroecological systems. This represents a major flaw of the publication which should preclude publication in its current form.
2. I do not agree with one of the central hypotheses of the study (which are in general not formalized in the text) but that is crucial for the actual interpretation of the data. This hypothesis is that similar community compositions are linked to similar environmental conditions. In this context, the authors completely overlook the long-standing debate regarding community assembly which can either be neutral or niche-based. Even if niche-based community assembly is assumed, this would require the use of metadata about community physico-chemical conditions in addition to the community composition to provide additional support to the clustering of the environments especially in relation to the inference of different lifestyle strategies for different taxa. The reviewer suggests that the authors' working hypotheses be explicitly stated in the article and references supporting the authors' opinions included.
3. The authors should make a concerted effort to provide concrete examples for their broad statements. The authors do make some attempts at this in L85-91 but should present examples at lower taxonomic rank and for more functions. The lack of compelling arguments for sweepingly broad claims should be rectified.

Minor reservations

L15: In which ecosystems do microbes not play a fundamental role?

L18-19: Some of these questions have been addressed by Muller et al., 2014, Nature Communications 5, 5603

L29: Use different term to “body”.

L48-49: What about sequences not represented in Silva, were these discarded? The authors should justify the 98 % cut-off. How do they know that the lifestyles of organisms within the 2 % will be congruent in different environments?

L72: environment(s)

L73-74: The null hypothesis need to be explained.

L77-79: It is impossible to judge this statement as the method section is cryptic at best and no actual results are presented to back up these claims

L253-264: This section lacks the necessary details: Which genomes were downloaded? How were they matched to the 16S rRNA data. How were the genes called, how were these annotated, how were functions inferred, etc.

1st Revision

Overall Response:

The authors would like to thank all reviewers for their comments. Many are critical suggestions that helped us further improve the analyses and the writing of the manuscript. We have addressed all comments to the best of our ability and the point-by-point responses are listed next to their corresponding comments below. We also would like to note that several minor edits were made to improve the manuscript's readability (listed at the bottom).

Reviewer #1 (Remarks to the Author):

The manuscript reports a massive analysis of the distribution of specialization of microbes. Specialization is measured as the number of 'environments' in which a given species is found. The main result is that generalists do have a much larger diversification rate than specialists. Based on this result, the authors develop the hypothesis that microbial evolution happens as a cycle of generalization and specialization.

I found the results very intriguing, and if they prove solid, I think it would make a very significant contribution to ecology and evolution, beyond microbial ecology. That said, I must admit that I am marginally qualified to judge the methodology that is employed in this study, and for most of what will follow I will assume that the technics are good. For me the most interesting part is the evolution cycle, which is quite contrary to convention with other organisms. The problem of the evolution of specialization has been investigated quite a lot, as well as the evolution of niches. Unfortunately, the authors do not place their results in this context and stay at the level of microbiology. I would encourage them to visit some of this literature and place their results in a wider context, asking if microbes are exceptional or coherent with other hypotheses (from what I see, these results are quite exceptional).

Response (R1C1): We would like to thank the reviewer for pointing out that the results of our study may reflect exceptional properties of microbes that could contribute to the broader discussion in ecology and evolution. We have added references to past studies in macroorganisms to the introduction (LINES 36-37), added discussions comparing our discoveries of higher speciation rates for generalists to past studies in macroorganisms (LINES 132-148), and compared the generalist-specialist evolutionary cycle to the adaptive radiation model (LINES 197-203).

While I am overall very positive about the paper, I have one general comment that might at the end be a major limitation preventing publication: the conclusions critically depend on how generalization / specialization is measured for each lineage. The authors cluster the usage of different environments, and then compute for each lineage the number of clusters it is found at. According to this approach, lineages that are qualified as generalists can't be otherwise, they are found at several places. But the reverse is not necessarily true, as a species could be qualified as specialist only because it is rare. The odds of a rare lineage be found at several locations are much smaller, up to the point where a lineage observed only twice could not be found at more

than two locations. Consequently, there is no guarantee for now that lineages qualified as generalists are true generalists, even with the null model analysis performed.

There is at least one way to check if this is problematic or not: the authors should look at the relationship between the number of times of a given species is observed against the number of habitats it is found at. If the relationship is triangular, then there is a big problem. In such case, the analysis could not discriminate between the effect of the strategy and of abundance.

Response (R1C2): We completely agree with the reviewer that rare species could be erroneously classified as specialists just because they were detected in only one sample. In accordance with the reviewer's suggestion, we have estimated the species abundances using 16S fragment read count (Methods LINES 244-262) and showed that in our dataset, species that were detected in only one environment were not enriched with rare species (Supplemental Figure S5 and LINES 179-181). Therefore, we do not expect the misclassification of rare species as specialists to be frequent.

I could see two ways to bypass this problem:

- first, instead of using binary descriptors of the strategy, the authors could use a continuous measure of specialization. There are tons of them, but one that is easy to perform is simply the Shannon entropy computed on the frequency of different environments for a given taxa. This approach would not completely solve the problem (would be hard to compute on rare species), but at least it would account for the fact that a species might have been found at 5 different environments, but 99% of its presence is at a single environment (this species would then qualify as a specialist, despite being found everywhere).

- The other option is to perform the opposite of rarefaction analysis and compute the asymptotic specialization as sequences are removed. This way, the measure would standardize the number of occurrences.

Response (R1C3): To test the reviewer's concern that some generalists may be better classified as specialists if they are present in high abundance in only one or a few environments, we used the species abundance data to show that the vast majority of classified generalists are highly abundant even in their 3rd most or 4th most dominant environment clusters (Supplemental Figure S5 and LINES 181-185). In combination with our reply to the reviewer's earlier point (R1C2), we believe the reviewer's concern regarding our classification of generalists and specialists is now properly addressed.

Other than type, I would propose extra testable predictions to test the evolution cycle hypothesis:

- If the hypothesis is right, then two sister species would be less likely to have the same strategy (generalist/specialist) than two species picked at random (also called overdispersion of strategies)

- The distribution of preferred environments should however be clustered (sister species are more likely to be found at the same environment)

Response (R1C4): We appreciate the reviewer's thoughts. However, the situation is not so straightforward and neither hypothesis could be positively confirmed with our dataset. The main reason is likely because we do not have high species coverage due to having to map 16S rRNA fragments to SILVA database and to the pre-constructed phylogenetic tree obtained from the Living Tree Project. Without extensive coverage of specialist species together and their most immediate generalist ancestors, it would be difficult to prove the two hypotheses. Furthermore, our model predicts that once a generalist lineage expanded into multiple environments, most if not all of its descendants would evolve into specialists. Depending on the amount of time since the initial expansion, we could either see sister species being all generalists, a mixture of generalists and specialists, or all specialists. Additionally, the broad scale of environment annotations in our dataset could render the pattern of sister species sharing the same environment indistinguishable from that between distant species.

I have very few minor points, the manuscript was well written and prepared. I really enjoyed reading it.

- Definition of specialisation should come early in the manuscript

Response (R1C5): We have added the definition of specialization in microbes to the introduction (LINES 31-32).

- L22 : replace 'survival' by 'persistence'

- L43 : it would be useful to give some ideas of the types of environments that are documented in the main text

Response (R1C6): Suggested edit has been made. In addition, examples of environments being studied have been added to the main text (LINES 44-45).

- L45 : replace 'predicted' by collected

- L46: what is the fraction of singletons ?

Response (R1C7): We have rephrased the sentence to avoid using 'predicted' (LINES 47-48). The fraction of singletons has been added (LINE 229).

- Some discussion on the sensitivity of the analysis to the definition of what constitutes a species should be provided.

Response (R1C8): Discussion on the sensitivity of the analysis to the definition of a microbial species has been added (LINES 163-172). Supplemental Figure S1 and S4 showed that the findings are robust to changes in the sequence identity threshold for defining operational taxonomic units (98% and 95%).

I signed my evaluation

Dominique Gravel

Reviewer #2 (Remarks to the Author):

The paper investigates the dynamics of diversification of microbial systems across a range of environments. The authors identify some unexpected patterns in the evolutionary dynamics of specialist and generalist microbial species. Overall this study is quite thorough, the results are extremely interesting and important. They also appear to be solid, and the writing style is good.

- What I find missing are any links to speciation/adaptive radiation literature/thinking in evolutionary biology. This is a shame because this paper has a potential to greatly contribute to ongoing discussions of general patterns of biological diversification in evolutionary biology (e.g., Gavrillets and Losos 2009, Science 323:732-737).

Response (R2C1): We have added more discussions linking our findings to past studies of generalist and specialist macroorganisms (LINES 132-148) and to adaptive radiation literatures (LINES 197-203).

- Birand et al (2012, Am. Nat. 79:1-21) studied speciation dynamics within a generalist-specialist framework. Some of their theoretical predictions should be contrasted with the authors' results.

Response (R2C2): This discussion has been added (LINES 141-148).

- Also, the title doesn't really reflect what this paper is about.

Response (R2C3): We have updated the title to “Generalist species drive microbial dispersion and evolution”.

Reviewer #3 (Remarks to the Author):

The manuscript by Sriswasdi et al. describes a meta-analysis of existing 16S rRNA data from different environments with the aim of inferring survival strategy (generalists vs. specialists) and linking these to different evolutionary patterns. The inference of microbial lifestyle strategies from omics data is an emerging area in microbial ecology and so far rather limited work has been done on translating these concepts from macroecology into microecology (but see Muller et al., 2014, Nature Communications 5, 5603). Although this paper tackles important questions, I am presently unconvinced about the approach given major flaws in the author's interpretation of what generalists and specialists represent. Please find my detailed comments below.

Response (R3C1): We are grateful for the reviewer's critical comments on potential flaws in our analyses. These prompted us to incorporate additional analyses and discussion that should address all reviewer's concerns and substantially strengthen the manuscript.

Major concerns

1. The use of the terms generalist and specialist as well as the linked concepts are used incorrectly. A generalist is able to thrive in a wide variety of environmental conditions and can utilise a variety of different resources – a generalist has a broad ecological niche. In contrast, a specialist can only thrive in a narrow range of environmental conditions or has a limited diet. As the

authors do not take into account population sizes, environmental conditions in the different environments they are looking at, different resource usage patterns by the different species, etc. they are unable to discern between generalists and specialists. In fact, from their analyses, the authors are able to differentiate between cosmopolitan and endemic species but are clearly unable to infer survival strategy. This misinterpretation of the data results in all kinds of issues for example the much higher speciation rate for “generalists” which is completely counter to what has been described for macroecological systems. This represents a major flaw of the publication which should preclude publication in its current form.

Response (R3C2): We understand that the terms generalist and specialist are used to distinguish species based on their habitat conditions and resource utilizations while the terms cosmopolitan and endemic are used to distinguish species based more on their geographic, physical distributions. To clarify our intention, we have included the definitions of generalist and specialist to the introduction (LINES 31-32). In our dataset, environments are annotated with terms that reflect more of their conditions than their location. For example, ‘soil’ environment consists of all samples from soil, regardless of geographical locations, and ‘human lung’ environment consists of all samples from human lungs, regardless of individuals. Therefore, we believe that the terms generalist and specialist could be extended to describe species that are present in multiple environments or few environments in this dataset. Furthermore, the fact that our approach effectively merge environments with related annotations (e.g. soil and rhizosphere in one cluster and human lung, nasal pharyngeal, and oral in another cluster) and similar species abundance profiles (LINES 70-74, Supplemental Figure S2), suggests that each cluster represents a habitat type and that our species classification scheme would distinguish between generalists and specialists.

In regard to the comment about our finding of higher speciation rate for generalist, we have added discussion connecting our finding to past studies of macroorganisms which revealed that different systems and methodologies could yield distinct conclusions (LINES 132-148).

2. I do not agree with one of the central hypotheses of the study (which are in general not formalized in the text) but that is crucial for the actual interpretation of the data. This hypothesis is that similar community compositions are linked to similar environmental conditions. In this context, the authors completely overlook the long-standing debate regarding community assembly which can either be neutral or niche-based. Even if niche-based community assembly is assumed, this would require the use of metadata about community physico-chemical conditions in addition to the community composition to provide additional support to the clustering of the environments especially in relation to the inference of different lifestyle strategies for different taxa. The reviewer suggests that the authors’ working hypotheses be explicitly stated in the article and references supporting the authors’ opinions included.

Response (R3C3): Upon this comment, we expanded the discussion on the results of environment clustering to stress the fact that each of the 11 clusters actually contain related environments, such as soil and rhizosphere in one cluster and lung, nasal pharyngeal, and oral in another cluster. In microbial ecology, it is well established that microbes are not neutrally distributed across environments at a broad scale. For example, microbial ecology textbooks

frequently use terms such as “soil microbes” and “gut microbes” and argue the similarities of their community compositions. Accordingly, we have stated our hypothesis and its prerequisites (microbial ecology on a broad scale of environments) and provided supporting reference (LINES 73-74).

3. The authors should make a concerted effort to provide concrete examples for their broad statements. The authors do make some attempts at this in L85-91 but should present examples at lower taxonomic rank and for more functions. The lack of compelling arguments for sweepingly broad claims should be rectified.

Response (R3C4): We have expanded the discussion to include more examples (LINES 98-112). The taxonomic groups most significantly biased toward being generalists or specialists in our dataset were consistent with previous reports in microbial ecology studies. However, regarding the reviewer’s request for more functional analyses, we wonder if that should be presented as part of a separate manuscript. This is because the focus of the current work is more on ecological aspects than on molecular mechanisms. Furthermore, as the molecular mechanisms that underlie the evolution of generalists and specialists would differ across microbial clades, they cannot be briefly discussed and may distract from the main message. We believe that the newly added examples and discussion already accomplished the goal of validating the meta-analysis approach of community sequencing datasets. In accordance with this, we have also added a statement that different mechanisms (other than flagellar) may be responsible for the enrichment of generalists or specialists in different microbial clades.

Minor reservations

L15: In which ecosystems do microbes not play a fundamental role?

L18-19: Some of these questions have been addressed by Muller et al., 2014, Nature Communications 5, 5603

Response (R3C5): We have revised the abstract and introduction to clarify that microbes are important in all ecosystems (LINES 16 and 28) and acknowledged Muller et al., 2014, Nature Communications (LINES 17-19, and 39).

L29: Use different term to “body”.

L48-49: What about sequences not represented in Silva, were these discarded? The authors should justify the 98 % cut-off. How do they know that the lifestyles of organisms within the 2 % will be congruent in different environments?

L72: environment(s)

Response (R3C6): A statement that sequences not represented in SILVA were removed has been added (LINE 52). Regarding the sequence identity cutoff issue, we very much understand the reviewer’s concern; however, we also understand that any threshold cannot guarantee that lifestyles of organisms within that threshold are completely congruent. In this study, we chose a 98% threshold that is more stringent than the commonly used 97% for defining microbial species.

We have added discussion on this topic to the main text (LINES 64-67). Other suggested edits have also been made.

L73-74: The null hypothesis need to be explained.

Response (R3C7): The null hypothesis is now explained (LINES 85-86).

L77-79: It is impossible to judge this statement as the method section is cryptic at best and no actual results are presented to back up these claims

L253-264: This section lacks the necessary details: Which genomes were downloaded? How were they matched to the 16S rRNA data. How were the genes called, how were these annotated, how were functions inferred, etc.

Response (R3C8): We are sorry for not properly explaining the method involved in obtaining and analysis of genome and proteome statistics. The method section has been extensively expanded (LINES 341-364) and a brief explanation to help interpreting the statement has been added to the main text (LINES 90-92).

Additional Revisions:

(LINES 57-58) Added a sentence to introduce the fact that we estimated species abundance data from 16S rRNA fragment read count.

(LINE 97) Corrected 'Mann-Whitney U test' to 'sign test'.

(LINES 130-131) Added a reference to support the finding of lower speciation rate in specialist microbes.

(LINES 149-151) Moved 'with significant and positive expansion rates (Figure 3B, the difference between speciation and extinction rates)' from the end of previous sentence.

(LINE 161) Removed 'that can be easily lost'.

2nd Round of Review

Reviewer #1 (Remarks to the Author):

I have looked carefully at the reply, with a particular attention to the method used for classification as either generalist/specialist. Note that reviewer 3 also criticized the method, although in a more qualitative way. I do not agree with his interpretation of the specialization concept, but we both agree that the conclusion of the study is critically contingent on the way it is measured.

The authors are fully aware of the classification problem, as evidenced by the extra discussion they put in the manuscript, and the additional figures S5a-b. This is already a good step forward.

They argue that the classification is robust based on two additional results they provide at fig S5c-d.

Honestly, explanations for Fig S5d are not sufficient, I simply can't interpret that figure. Further, it looks to me it focuses on generalists species, while the problem is much more about the classification of specialists.

And I don't think the analysis for Fig. S5c is the right way to go. There are too many manipulations of the data behind this figure (including the identification of clusters, transformation of abundances etc....), again precluding its interpretation. Further, it is not a direct test of the effect of abundance on the classification.

There is a much simpler analysis to perform, which would be far more convincing. I'll be straightforward here : I will oppose to the publication of this study if this analysis is not performed.

Each sequence is linked to an environment it has been collected, and eventually to a cluster. The main question is : is a sequence cluster found on more or less environmental clusters than by chance alone, given the number of times it has been observed ? The answer needs to be insensitive to the number of times it has been observed. If the answer is 'more' then it should be classified as a generalist, if it is 'less' then a specialist, and if it matches the random expectation then it is neither of them. To answer this question, the authors simply have to attribute randomly one environmental cluster for each sequence and then recompute the total number of environmental clusters it has been observed at. The ratio of the observed number of environmental clusters relative to the random expectation is a fairly strong measure of specialization that is independent of abundance.

Following this reclassification, the authors have two options : either they compare the results to the previous classification and if they prove solid then they keep it, otherwise they need to re-run their analysis.

Further, there is another suggestion that I've made but the authors never considered (they simply do not respond to it) : they could compute the Shannon entropy for each sequence cluster across the different environments. This measure of specialization would also be independent of total abundance and easily comparable across sequence clusters.

Many readers will emphasize this critical limitation of the study. The authors definitely need to perform a direct, easy to interpret and convincing analysis that their classification is appropriate.

Reviewer #2 (Remarks to the Author):

Thank you for the thorough revision

Reviewer #3 (Remarks to the Author):

The authors have performed a thorough revision of their original manuscript. Although, I do still not entirely agree with the authors' use of the terms "generalist" and "specialist", I am aware that other individuals are also using these terms along the same lines as the authors and this is an academic argument one can have but which should not preclude the publication of this important work.

2nd Revision

Overall Response:

The authors would like to once again thank all reviewers for their considerations and to offer a sincere apology to the first reviewer for our failure to properly respond to all of his suggestions. We have addressed the new comments and revisited some old ones to the best of our ability. Please find point-by-point responses listed next to their corresponding comments below. We have also listed additional minor edits toward the end of our response.

Reviewer #1 (Remarks to the Author):

I have looked carefully at the reply, with a particular attention to the method used for classification as either generalist/specialist. Note that reviewer 3 also criticized the method, although in a more qualitative way. I do not agree with his interpretation of the specialization concept, but we both agree that the conclusion of the study is critically contingent on the way it is measured.

The authors are fully aware of the classification problem, as evidenced by the extra discussion they put in the manuscript, and the additional figures S5a-b. This is already a good step forward.

They argue that the classification is robust based on two additional results they provide at fig S5c-d.

Honestly, explanations for Fig S5d are not sufficient, I simply can't interpret that figure. Further, it looks to me it focuses on generalists species, while the problem is much more about the classification of specialists.

And I don't think the analysis for Fig. S5c is the right way to go. There are too many manipulations of the data behind this figure (including the identification of clusters, transformation of abundances etc....), again precluding its interpretation. Further, it is not a direct test of the effect of abundance on the classification.

Response (R1C1): We understand the reviewers' concern and have now improved the investigation of classification problem according to the reviewer's suggestions for using Shannon entropy (LINES 179-184 and new Figure S5C) and a permutation test (LINES 185-197). Figure S5C-D have been replaced with new data which show that removal of classified species that are most susceptible to misclassification has little impact on the estimated evolutionary characteristics of the generalists and specialists. For specialists, we removed classified species with the lowest relative abundances because rare species are expected to be particularly susceptible to misclassification (Figure S5A). For generalists, we computed the Shannon entropy for the distribution of their relative abundances across 11 environment clusters and removed those with low entropy (Figure S5B). These revisions in conjunction with additional responses below should constitute solid evidence that our findings are robust.

There is a much simpler analysis to perform, which would be far more convincing. I'll be straightforward here : I will oppose to the publication of this study if this analysis is not performed.

Each sequence is linked to an environment it has been collected, and eventually to a cluster. The main question is : is a sequence cluster found on more or less environmental clusters than by chance alone, given the number of times it has been observed ? The answer needs to be insensitive to the number of times it has been observed. If the answer is 'more' then it should be classified as a generalist, if it is 'less' then a specialist, and if it matches the random expectation then it is neither of them. To answer this question, the authors simply have to attribute randomly one environmental cluster for each sequence and then recompute the total number of environmental clusters it has been observed at. The ratio of the observed number of environmental clusters relative to the random expectation is a fairly strong measure of specialization that is independent of abundance.

Following this reclassification, the authors have two options : either they compare the results to the previous classification and if they prove solid then they keep it, otherwise they need to re-run their analysis

Response (R1C2): We have performed the permutation procedure as suggested. The corresponding revisions were added to the main text (LINES 185-197) and the method section (LINES 404-414). The estimated evolutionary characteristics for generalist and specialist exhibited consistent patterns across the two methods (Figure S3 and Figure S6).

Further, there is another suggestion that I've made but the authors never considered (they simply do not respond to it) : they could compute the Shannon entropy for each sequence cluster across the different environments. This measure of specialization would also be independent of total abundance and easily comparable across sequence clusters.

Response (R1C3): Please let us again offer our apology for not properly responding to this suggestion previously. We have now incorporated Shannon entropy as a way to determine classified generalists that may result from misclassification of non-generalist species (those with low entropy, LINES 179-184 and Figure S5C).

Many readers will emphasize this critical limitation of the study. The authors definitely need to perform a direct, easy to interpret and convincing analysis that their classification is appropriate.

Response (R1C4): We completely agree with the reviewer and we believe that the new revisions in combination with old ones would constitute convincing evidences that our main findings are robust to variations in classification and analysis parameters.

Reviewer #2 (Remarks to the Author):

Thank you for the thorough revision

Reviewer #3 (Remarks to the Author):

The authors have performed a thorough revision of their original manuscript. Although, I do still not entirely agree with the authors' use of the terms "generalist" and "specialist", I am aware that other individuals are also using these terms along the same lines as the authors and this is an academic argument one can have but which should not preclude the publication of this important work.

Response: Thank you very much for your consideration and understanding.

Additional Minor Revisions:

(LINES 6-7) The first author has moved to a new position in June 2017 and conducted this revision there.

(LINE 82) Previously listed numbers of generalist and specialist species correspond to the number of individual SILVA ID. We have updated the numbers so that they reflect the number of species (or clusters of SILVA IDs) instead. Table S4 already contains both levels of information.

(LINE 106) Changed “motility” to “flagellar motility” for clarity.

(LINES 173-174) Changed “they may be derived at” to “similar findings may be found at” to improve readability.

(LINE 176) Added panels A and B to Figure S5 reference for clarity.

(LINE 276) Corrected figure reference from Figure S3 to Figure S2.

(LINE 301) Same as the correction for LINE 82.

(LINE 402) Corrected “is consider” to “is considered”

(LINE 520) Added Sohta Ishikawa to the acknowledgement.

Final Review

Reviewer #1 (Remarks to the Author):

I would like to apologize to the authors for taking so much time for performing my evaluation of the reply. I was out for field work, followed by trips and many other activities preventing me for doing it. I feel terribly sorry for this long delay.

That said, it was a pleasure to look at the revisions and see that the results are robust to the classification made. The authors did consider my comments seriously and performed appropriate extra analyses. As far as I could see, the methods are robust and the conclusions are appropriate. I am therefore happy to recommend publication of this great study.